# The Role of Tregs in the Tumor Microenvironment

**DOI:** 10.3390/biomedicines13051173

**Published:** 2025-05-11

**Authors:** Yohei Sato

**Affiliations:** 1Laboratory of Immune Cell Therapy, Project Research Unit, The Jikei University School of Medicine, Tokyo 105-8461, Japan; yoheisato@jikei.ac.jp; Tel.: +81-3-3433-1111 (ext. 2430); 2Core Research Facilities, Research Center for Medical Sciences, The Jikei University School of Medicine, Tokyo 105-8461, Japan; 3Immunology and Allergy Research Unit, Division of Otorhinolaryngology Head & Neck Surgery, Faculty of Medicine, University of Fukui, Fukui 910-1193, Japan

**Keywords:** FOXP3, regulatory T cells, immune regulation, tolerance, tumor-infiltrating lymphocyte, cancer

## Abstract

The tumor microenvironment (TME) is a unique ecosystem that surrounds tumor tissues. The TME is composed of extracellular matrix, immune cells, blood vessels, stromal cells, and fibroblasts. These environments enhance cancer development, progression, and metastasis. Recent success in immune checkpoint blockade also supports the importance of the TME and immune cells residing in the tumor niche. Although the TME can be identified in almost all cancer types, the role of the TME may not be similar among different cancer types. Regulatory T cells (Tregs) play a pivotal role in immune homeostasis and are frequently found in the TME. Owing to their suppressive function, Tregs are often considered unfavorable factors that allow the immune escape of cancer cells. However, the presence of Tregs is not always linked to an unfavorable phenotype, which can be explained by the heterogeneity and plasticity of Tregs. In this review, the current understanding of the role of Tregs in TME is addressed for each cancer cell type. Moreover, recently a therapeutic approach targeting Tregs infiltrating in the TME has been developed including drug antibody conjugate, immunotoxin, and FOXP3 inhibiting peptide. Thus, understanding the role of Tregs in the TME may lead to the development of novel therapies that directly target the TME.

## 1. Introduction

The tumor microenvironment (TME) is a unique ecosystem surrounding tumor tissues that helps escape immune recognition and elimination by the adaptive immune system [1]. The components of the TME are tumor-specific but typically comprise the extracellular matrix, immune cells, blood vessels, stromal cells, and fibroblasts [2,3]. Cancer cells utilize TME. Among various immune cells, tumor-infiltrating lymphocytes (TIL) and tumor-associated macrophages (TAM) contribute to the generation of the tumor niche [4,5,6]. Immune cells residing in tissues have been identified in various organs and have adapted to these tissues [7]. Recent breakthroughs in cancer immunotherapy have highlighted the importance of local immune reactions that occur or are influenced by the TME [8]. Tregs express inhibitory molecules, including PD-1, PD-L1, and CTLA-4; therefore, checkpoint blockade can potentially activate Tregs [9]. In addition to the adaptive immune system, the role of the innate immune system has been indicated in tumor immunity, and crosstalk between the innate and adaptive immune systems can orchestrate this process [10]. The importance of TME in tumor metastasis has also been recognized [11,12]. Therefore, understanding TME is critical for developing novel therapeutic approaches [13]. A checkpoint blockade targets the TME by modulating the immune cells residing in the TME. Recent studies have targeted the ECM, blood vessels, stromal cells, and fibroblasts. Multiple researchers have described the complexity and heterogeneity of therapeutic approaches targeting the TME [14].

This review discusses the role of regulatory T cells (Tregs) in the TME and recent clinical and translational investigations on antidrug conjugates, immunotoxins, and FOXP3 inhibitory peptides. Understanding the role of Tregs in the TME may contribute to TME-directed therapies.

## 2. Biological Role of Tregs in the Local Immune System

Tregs are a unique cell population that contributes to immune homeostasis through their suppressive function [15]. FOXP3 is a master transcription factor that regulates Treg differentiation and phenotype. The phenotype and suppressive functions of Tregs are controlled mainly by FOXP3 expression [16,17]. Mutations in FOXP3 in humans result in IPEX syndrome, whereas mutations in Foxp3 cause scurfy in mice, both of which are characterized by lethal autoimmunity owing to a lack of functional Tregs [18]. Genetic mutations and the instability or dysfunction of Tregs have been reported in multiple autoimmune, allergic, neurological diseases, and cancer [19]. It is commonly believed that FOXP3 expression and suppressive functions are highly correlated. FOXP3 drives the expression of co-inhibitory molecules, including CTLA-4. Here, we address the basic phenotype and function of Tregs and their plasticity and heterogeneity.

### 2.1. Phenotype and Function of Tregs

In mice, Tregs were initially identified as CD4+ and CD25+ cells [20]. FOXP3 has been identified as the master regulator [16]. CD4(+)CD25(+)CD127(low/−) cells express FOXP3, and FOXP3+ Tregs can be isolated without FOXP3 staining [21]. In humans, Tregs are known to express FOXP3 delta-2 isoform lacking exon 2. The role of FOXP3 delta-2 isoform is not fully confirmed in the context of TME. In addition to “bona fide” Treg markers, Tregs express CTLA-4, GARP, GITR, and IKZF2, which are known FOXP3 binding targets (Table 1) [22]. Therefore, high and stable FOXP3 expression in Tregs is essential for their phenotype and suppressive function.

Other markers (OX40, CD69, TIGIT, ICOS, Lag-3, PD-1, CD39, and CD73) were also included in the phenotypic assessment (Figure 1A) [23]. However, several other molecules (CD69, TIGIT, and ICOS) are also expressed by effector T cells upon activation and may not be specific to Tregs. An in vitro suppression assay was conducted to assess suppressive function [24,25]. In addition to phenotyping, the stability and function of Tregs have been assessed in vitro. Notably, most Treg-related molecules, including CTLA-4, GARP, GITR, TIGIT, ICOS, and PD-1, have been utilized as therapeutic targets in cancer immunotherapy. Therefore, understanding the Treg phenotype in cancer TME could provide information on potential targets for cancer immunotherapy.

### 2.2. Plasticity and Heterogeneity of Tregs

Tregs exhibit plasticity, particularly when affected by inflammation and proinflammatory cytokines [26,27,28]. In acquired autoimmune diseases, the loss of Treg identity can contribute to disease progression [29]. Tregs express IL-1R and IL-6R, which are stimulated by IL-1 and IL-6 cytokines, and may lose FOXP3 expression [30]. Supporting these findings, genetic inhibition of IL-1R and IL-6R enhanced Treg stability in vitro [31]. These observations provide evidence of Treg plasticity, which may be necessary for maintaining immune homeostasis according to the external environment.

Other factors also contribute to Treg/Th17 cell differentiation, which may lead to inflammation and autoimmunity [32,33,34]. In addition to their plasticity, Tregs have subpopulations similar to those of effector T cells [35,36]. It is still not yet fully discovered when and how these subpopulations were developed and contributed to heterogeneity during differentiation. This plasticity and heterogeneity provide Tregs temporal diversity (Figure 1B). The local metabolic status also contributes to Treg heterogeneity [27]. It has been speculated that the phenotype and identity of Tregs may be heterogeneous in each cancer tissue and may be affected by infection and inflammation.

### 2.3. Tissue-Resident Tregs

Tregs have been identified in local non-lymphoid tissues, including adipose tissues, muscles, skin, and the gastrointestinal tract [37]. Tissue-resident T cells contribute to immune reactions in local tissues [38,39,40]. Barrier tissues, including the respiratory tract and skin, contain resident memory cells, primarily for a rapid response to known antigens (Figure 1C). Similarly to TRM, tissue-resident Tregs have been proposed but not readily identified. Recently, single-cell sequencing has allowed for the identification of tissue-resident Tregs; however, the clonality of tissue-resident Tregs is similar among different tissues, and whether these Tregs are strictly tissue-resident remains unclear [41,42,43]. Tissue-resident T cells and Tregs contribute to cancer initiation; however, the exact roles of these newly identified T cells in carcinogenesis remain unclear. Tregs’ heterogeneity, plasticity, and spatiotemporal diversity can influence local immune homeostasis and may contribute to cancer progression and metastasis. Further studies are needed to confirm the phenotypes and functions of tissue-resident Tregs under physiological and pathological conditions.

## 3. Phenotype and Function of Tregs Isolated from TME

Tregs reside in the TME of multiple tumor types (Table 2). Due to their immunosuppressive effects, their presence in the TME and/or tumor tissues is detrimental. However, in some cancer types, Tregs in the TME could be associated with favorable outcomes (Figure 2) [44]. These complex clinical observations indicate that the biological roles of Tregs differ across different types of cancers. Therefore, determining whether Tregs suppress tumor immunity or enhance tumor elimination is challenging. The current interpretation of tumor-infiltrating Tregs is discussed according to specific cancer types.

### 3.1. Colorectal Cancer

Most immune cells reside in the gastrointestinal tract; hence, the presence of immune cells in the tumor niche was presumed (Figure 3) [45]. Although Tregs are detected in the gastrointestinal tract under physiological conditions, they have also been identified in colon cancer [46]. Tregs isolated from colon cancer cells exhibit effector phenotypes [47]. Treg depletion or inhibition can enhance the antitumor activity [48,49]. The blockade of checkpoints, including CTLA-4 and PD-1, targets effector T cells and Tregs in the TME [50]. However, Treg infiltration is associated with a better prognosis. Tregs, characterized by reduced FOXP3 expression, may not be suppressive and enhance local immune reactions [51]. Treg plasticity may be beneficial for tumor immunity. According to these data, Tregs are not entirely responsible for cancer progression, and Treg ablation may not be helpful in colon cancer. However, enhancing tumor immunity could benefit patients with colon cancer.

### 3.2. Lung Cancer

Resident memory T cells have also been identified in the respiratory tract [39,40]. Like the gastrointestinal tract, Tregs have been identified in the lungs under physiological conditions. The involvement of Tregs has also been identified in other lung diseases, including acute lung injury and infections (Figure 4) [52,53]. It has been speculated that Tregs control immune homeostasis in the lungs under both physiological and pathological conditions.

Tregs have also been identified in lung cancer [54]. Treg infiltration in lung cancer may result in divergent phenotypes [55]. COX-2 expression may be associated with the long-term prognosis in non-small cell lung cancer [56]. Unlike colorectal cancer, the literature on Treg infiltration resulting in favorable outcomes is unavailable. Thus, Treg infiltration may result in immune escape, contributing to tumor development and relapse. The alteration of the TGF-beta signaling pathway in the tumor niche may be negatively associated with Treg infiltration [57]. To support these results, selective depletion of CCR8+ Tregs was beneficial for lung cancer in a mouse model [58]. The current clinical and translational evidence suggests the beneficial effects of Treg depletion in lung cancer.

### 3.3. Breast Cancer

Unlike the gastrointestinal and respiratory tracts, which are considered barrier tissues exposed to foreign antigens, breast tissue may not contain many immune cells under physiological conditions. However, inflammation is also associated with breast cancer progression [59]. Tregs isolated from breast cancer cells exhibit a unique phenotype including CCR8 expression [60]. Recently, immunotherapy such as PD-1/PD-L1 immune checkpoint blockade has been used to manage triple-negative breast cancer, suggesting the involvement of Tregs in the TME [61]. In addition to their divergent phenotypes such as co-inhibitory molecule expression, Tregs in breast cancer may contribute to disease progression (Figure 5) [62]. Moreover, the interaction between Tregs in PBMC and the cancer niche may cause disease progression and relapse [63]. Interestingly, the unique metabolic and transcriptomic profile of breast cancer may be associated with Tregs in breast cancer tissues according to the cancer subtype [64]. Indeed, Treg ablation was beneficial and enhanced radiotherapy in the mouse model [65]. Alterations in the TME, including the presence of Tregs, are associated with breast cancer progression [66]. These results indicate the detrimental effects of Tregs in breast cancer and suggest that it may be beneficial to deplete or suppress Tregs in the TME of breast cancer cells.

### 3.4. Pancreatic Cancer

Although the pancreas is not an immune organ, there is much evidence suggesting that immune cells play a central role in inflammation and cancer development [67]. Pancreatic Tregs have been studied more frequently in patients with type 1 diabetes [68,69,70,71]. Pro-insulin- or GAD-specific Tregs that protect islets from autoimmunity, and the loss or dysfunction of antigen-specific Tregs, may contribute to the development of type 1 diabetes. Adoptive Treg transfer has been conducted in multiple clinical trials based on Treg dysfunction in T1D [72]. In contrast, pancreatic cancer, called a “cold tumor”, is characterized by minor immune cell infiltration [73]. TME has also been identified in pancreatic cancer [74,75]. Tregs have also been identified in pancreatic cancer (Figure 6) [76]. A recent study also suggested a functional role for Tregs in the TME of pancreatic cancer [77]. Unlike in other cancer types, Treg depletion has shown detrimental effects [78]. Thus, the TME is controlled and maintained through different mechanisms in different cancer types. Immunotherapy, such as PD-1 and CTLA-4 checkpoint blockade, has been indicated for pancreatic cancer management; however, its efficacy has not been fully confirmed, partially because of disease severity [79]. The role of Tregs in pancreatic cancer has not been fully elucidated. Thus, further clinical and translational studies of Tregs in pancreatic cancer are warranted to validate their possible role in cancer progression.

### 3.5. Hepatocellular Carcinoma

Although the human liver is not usually considered an immune organ, the presence of various immune cells has been well-documented [80]. As previously suggested, a recent study identified multiple immune cells, including Tregs, in the liver [81]. Both the innate and adaptive immune cells are activated [82,83]. Tregs reside in the liver tissue, and liver Treg dysfunction such as IL-17 production is associated with fibrosis [84]. Tregs have been speculated to reside in the liver under healthy conditions and regulate immune responses and regeneration, including fibrosis. TME has also been identified in hepatocellular carcinoma [85,86]. The association between cancer and Treg infiltration is not evident compared with other types of cancer [87]. Chronic HBV and HCV infections can contribute to the development of hepatocellular cancer; it has been suggested that HBV/HCV potentially utilizes Tregs to escape the immune system, similar to other cancer cell types [88]. Their association with chronic infection may complicate determining the role of Tregs in the TME independent of HBV/HCV infection. The lack of clinical and translational studies in hepatocellular carcinoma focusing on the TME makes it challenging to consider immunotherapy and Treg ablation to enhance tumor immunity; there is little evidence for Treg ablation in hepatocellular carcinoma. Describing Tregs in hepatocellular carcinoma may provide insights into their therapeutic applications.

### 3.6. Brain Tumor

Unlike other tissues, the brain is considered an “immune-privileged site”; however, immune cells directly contribute to brain hemostasis [89]. The presence of Tregs in the brain has been confirmed and may contribute to neurological function [90]. Similarly to other cancer types, Treg infiltration can be an unfavorable prognostic factor for brain tumors (Figure 7) [91]. The brain tumor expresses various immunomodulatory molecules necessary to control immune response even in the “immune-privileged site”. PD-1/PD-L1 expression is also associated with clinical phenotypes [92]. Glioblastomas may suppress immune reactions owing to their strong IDO expression [93]. GITR targeting improves therapeutic efficacy [94]. These results suggest that brain tumors directly suppress the immune response through a similar suppressive mechanism through co-inhibitory molecules. This indicates that either immunotherapy or Treg ablation may enhance tumor immunity and improve the clinical outcome.

### 3.7. Melanoma

Skin tissue contains multiple immune cells, and various tissue-resident cells have been reported [95]. As expected, Tregs have been identified in the skin tissue, and their association with inflammation or allergies has been reported [96,97]. The TME has been identified in melanomas, similar to other cancers [98]. Tregs have also been identified in melanomas [99]. The success of PD-1 blockade in melanoma is a benchmark for immunotherapy. Therefore, immune regulation is expected to contribute to melanoma progression [100]. Concerning PD-1 blockade, the interaction between CD8+ T cells and Tregs is considered the key effector for tumor rejection [101]. Although the PD-1 blockade is curative in some patients, resistance to therapy must be investigated to improve treatment outcomes. The role of Tregs in melanoma progression remains controversial. The effects of Treg ablation should be confirmed in future studies [99].

### 3.8. Kidney and Urinary Tract Cancer

Unlike other tissues, tissue-resident immune cells, which are aseptic and may not contain immune cells, are typically absent from the urinary tract. However, the urinary tract may contain resident memory T cells; the presence of TRM has been confirmed after UTI [102]. The TME has been identified in renal cell carcinoma [103] and bladder cancer [104]. Tregs have also been identified in bladder cancer [105]. The roles of TME and Tregs in urinary tract cancers, including renal cell carcinoma, have not been extensively studied. Immunotherapy has been used for the management of bladder and prostate cancers [106]. CCR4 blockade inhibits Treg infiltration in bladder cancer in a canine model [107]. CXCR4 blockade inhibits Treg infiltration in renal carcinoma [108]. Although there is insufficient evidence, Treg ablation may benefit patients with kidney and urinary tract cancers.

### 3.9. Gynecological Cancer

The female reproductive system acts as a barrier to multiple immune cells [109]. Pregnancy and hormonal changes drastically alter systemic and local immunity [110,111]. These results suggested that the female reproductive tract is closely associated with immune cells under physiological conditions. The presence of tissue-resident immune cells, including T cells, B cells, and microphages, supports these results. TME has also been identified in ovarian cancer [112,113]. Moreover, single-cell sequencing has identified multiple components in the TME of ovarian cancer [114]. Tregs have also been identified in ovarian cancer [115]. Immune checkpoint blockade including PD-1 and CTLA-4 blockade has also been attempted in cervical cancer treatment [116]. The roles of the TME and Tregs in gynecological cancers are not fully understood, although immunotherapy has been attempted for several cancers including ovarian, cervical and uterine cancers. There is little evidence suggesting the effect of Treg ablation on gynecological cancers. Further studies exploring the role of Tregs in cancer development, progression, and metastasis are needed to develop a potential immunotherapeutic approach for female gynecological cancers.

### 3.10. Head and Neck Cancer

Head and neck cancers encompass multiple types of cancer that develop in the mouth, throat, nose, sinuses, and salivary glands. Because the head and neck contain multiple lymphoid tissues, including the lymph nodes and tonsils, a unique immune landscape has been described [117]. Immunotherapeutic approaches, including checkpoint blockade, have been investigated in various head and neck cancers [118]. Similarly to colon cancer, the infiltration of Tregs into cancer tissues is considered a favorable factor in head and neck cancers [119]. It was also reported that the increase in effector type (CD45RA-) FOXP3^high^Treg is negatively correlated with poor prognosis [120]. Another study indicated that CTLA-4+ Tregs might be associated with limited survival after anti-EGFR targeting by inhibiting natural killer (NK) cell function, which is critical for head and neck cancer. Based on the field’s controversy, concluding whether Tregs are favorable factors in head and neck cancer was impossible. Due to the heterogeneity of head and neck cancers, it may not be possible to determine the common effects of Tregs.

### 3.11. Soft Tissue Sarcoma

Soft tissue sarcoma is usually considered to be less immunologically malignant than other tumor types. However, the presence of Tregs in soft tissue sarcomas may be associated with poor survival [121]. Limited efficacy of immunotherapies such as PD-1 and PD-L1 blockade for soft tissue sarcoma has been reported [122]. The immune landscape of soft tissue sarcomas has not yet been thoroughly investigated, but the importance of immunomonitoring in soft tissue sarcomas has been suggested [123]. Few studies have described the biological functions of Tregs in soft tissue sarcomas. Due to the reduced immunogenicity, the association between Treg infiltration and clinical outcome is not anticipated; this needs to be confirmed in future clinical and translational studies.

### 3.12. Leukemia/Lymphoma

Similarly to other solid tumors, lymphomas have a TME [124,125]. Leukemia does not involve a tumor mass, and the TME has not been fully identified; however, several studies have indicated the involvement of Tregs in disease progression [126,127]. The functions of Tregs in acute and chronic GvHD have been previously studied [128]. The bone marrow niche is not considered a TME; however, leukemic stem cells may utilize the bone marrow as a TME, similar to other cancers [129]. Thus, the bone marrow niche may be a potential therapeutic target for leukemia [130]. Tregs support hematopoietic stem cell (HSC) engraftment and survival by harnessing the TME [131]. Similarly to HSC, Leukemic stem cells (LSCs) have also been supported by Treg [132]. Future studies investigating tumor/leukemic niches may provide evidence supporting the role of Treg ablation in hematological cancers.

## 4. Therapeutic Approach Targeting Treg in TME

Tregs are potential therapeutic targets for cancer immunotherapy [133]. Several therapeutic approaches have been proposed to target Tregs in TME [134]. Antibody–drug conjugates, immunotoxins, and peptides have frequently been investigated for targeting Tregs in the TME [135]. As discussed in the previous section, the role of Tregs in the TME may be tumor-dependent; their presence may be detrimental in most cancer types. It is essential to study the beneficial effects of Tregs on TME, as observed in specific cancer types, including colon and pancreatic cancers. The presence of Tregs in the TME has been extensively studied in cancers of the digestive, respiratory, and breast tissues, compared to gynecologic and urinary tract cancers. Understanding normal tissues and tissue-resident cells may contribute to evaluating and interpreting immune cells isolated from the TME. The current therapeutic approaches targeting Tregs in the TME are discussed in the following sections.

### 4.1. Antibody Drug Conjugate

Among the therapeutic candidates targeting the TME, antibody–drug conjugates are the most promising. Currently, more than 13 antibody–drug conjugate products have been approved by the FDA [136,137]. In a preclinical study, a CD25 antibody conjugated with pyrrolobenzodiazepine was shown to eliminate Tregs and enhance tumor immunity against lymphoma [138]. Its safety and efficacy have been tested in a Phase 1 clinical trial. A CD25 antibody conjugated to pyrrolobenzodiazepine may be possible for intratumoral Treg depletion.

### 4.2. Immunotoxin

Before the emergence of antibody–drug conjugates, immunotoxins targeting IL-2 and CD25 were studied to delete Tregs in vivo [139,140]. Similarly to CD25 antibody–drug conjugates, both IL-2 and CD25 immunotoxins successfully reduced Tregs in vivo and enhanced the T cell immune response suppressed by Tregs. Because antibody–drug conjugates and immunotoxins share the mechanism of action, CD25 targeting is currently the most efficient method of reducing Tregs.

### 4.3. Peptide

In addition to antibody–drug conjugates and immunotoxins, peptide–drug conjugates have also been investigated as Treg targets. CD28 targeting aptamer was combined with P60, a synthetic 15-mer FOXP3 inhibiting peptide; it was shown to inhibit Tregs [141]. P60 conjugated with CD25 targeting nanoliposome was shown to reduce intratumoral Tregs [142]. Although antibody–drug conjugates and immunotoxins directly eliminate Tregs, P60 inhibits FOXP3 and potentially reduces suppressive function. Although inhibited Tregs were not eliminated from the tissues, the peptide-based approach may have superior safety compared to other Treg-targeting strategies.

## 5. Future Investigations for Targeting Treg in TME

Early-phase clinical trials have investigated several therapeutic applications, such as the CD25 antibody conjugate. Immunotoxin and peptides showed similar efficacy in preclinical models. Tregs are depleted in vivo. However, monitoring the immune response after Treg depletion is essential to avoid unwanted immune responses such as autoimmunity. Unlike in mice, ablating Tregs in humans without adverse events is impossible. In particular, it is more challenging to ablate tumor-infiltrating Tregs than peripheral Tregs. Experiments in mice suggested that the blockade of the homing receptor (CXCR4) and chemokine receptor (CCR4/CCR8) may inhibit Treg infiltration and tumor homing. A more precise understanding of Tregs in TME may provide direct evidence for Treg ablation/depletion in the context of cancer immunotherapy.

Tregs and other immune cells, including effector T cells, NK cells, and macrophages, may be involved in the TME; however, their biological roles have not been fully determined. The roles of other non-immune cells, such as stromal cells, fibroblasts, and blood vessels, have not often been studied compared to those of immune cells. The interaction among immune, non-immune, and cancer/tumor cells must be investigated. The presence of heterogeneity in the cancer types, patient condition including immunity, other non-immune cells may contribute to the limited efficacy of TME-directed therapy in the current time point. A further understanding of both TME and immune cells including Tregs may benefit the discovery of TME-directed therapeutic approaches. However, this may not be possible due to patient sample limitations and ethical issues. Recent scRNA-seq or spatial transcriptomics studies may provide further insights with regard to immune profile of TMEs. The discovery of the crosstalk between cells in the TME may lead to novel therapeutic interventions.

## 6. Conclusions

The biological role of the TME and the functional properties of tumor-infiltrating Tregs have been vigorously investigated across various cancer types. Recent advances in tumor immunology, single-cell sequencing, and spatial transcriptomic analysis have identified tumor-infiltrating Tregs and proposed possible therapeutic candidates. The diversity of clinical backgrounds, TME, and Treg heterogeneity need to be considered, and the importance of clinical and translational studies is appreciated in the field. A precise understanding of TME and tumor-infiltrating Tregs may shed new light on cancer immunotherapy.

## Figures and Tables

**Figure 1 biomedicines-13-01173-f001:**
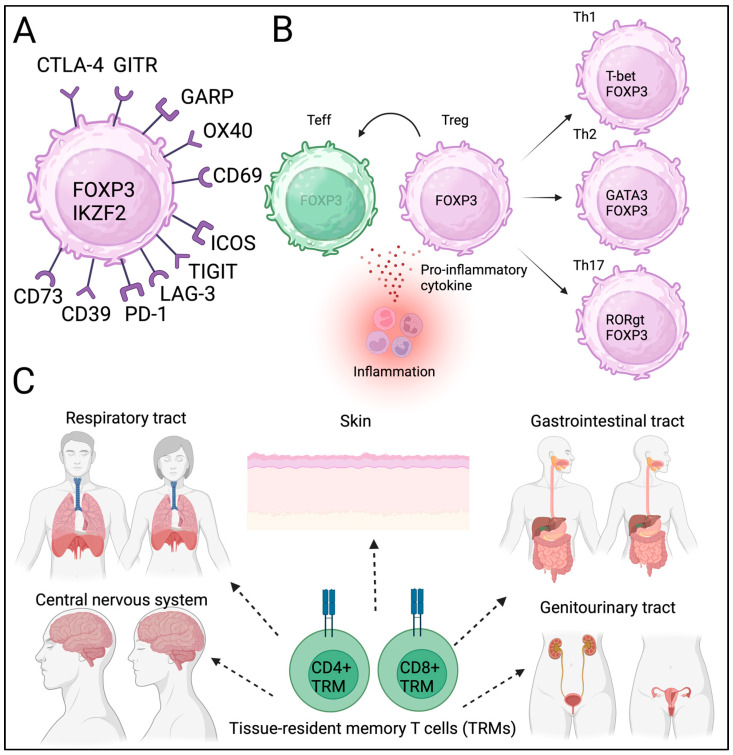
**Treg phenotype, plasticity, heterogeneity, and tissue residency.** (**A**) Core and additional Treg markers. (**B**) Plasticity and heterogeneity of Tregs. (**C**) Tissue resident memory T cells (TRM).

**Figure 2 biomedicines-13-01173-f002:**
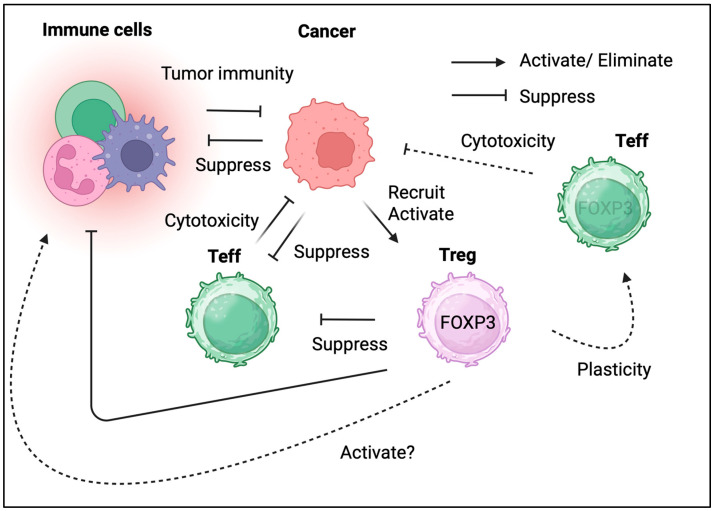
**The dual role of Tregs in cancer.** Tregs in TME both promoting tumor growth and potentially aiding in its elimination.

**Figure 3 biomedicines-13-01173-f003:**
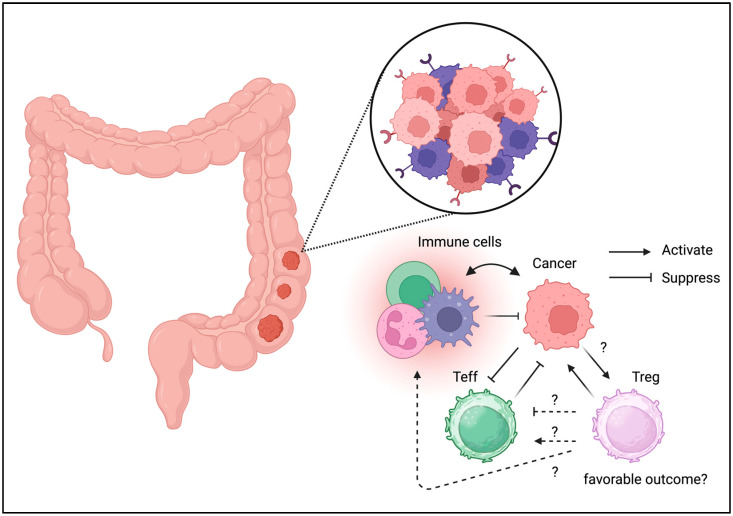
**Tregs can be detected in TME of colon cancer.** Unlike the other cancer types, Tregs may be associated with favorable outcome in colon cancer.

**Figure 4 biomedicines-13-01173-f004:**
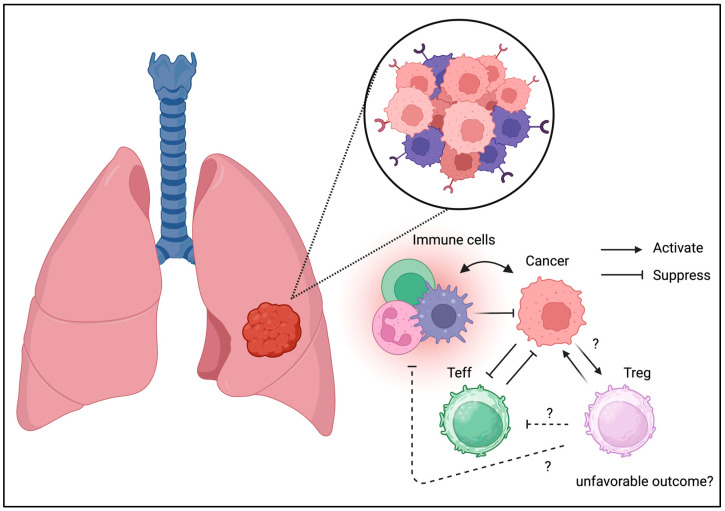
**Tregs can be detected in TME of lung cancer.** Similarly to the other cancer types, Tregs may be associated with unfavorable outcome in lung cancer.

**Figure 5 biomedicines-13-01173-f005:**
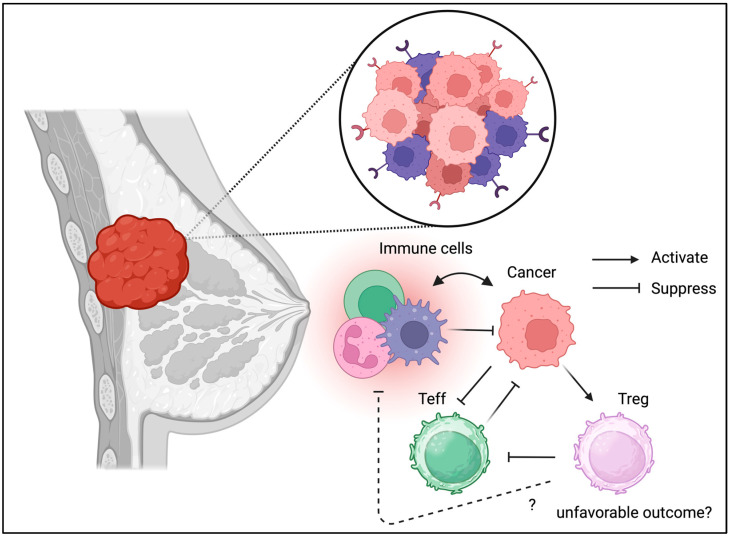
**Tregs can be detected in TME of breast cancer.** Similarly to the other cancer Types, Tregs may be associated with unfavorable outcome in breast cancer.

**Figure 6 biomedicines-13-01173-f006:**
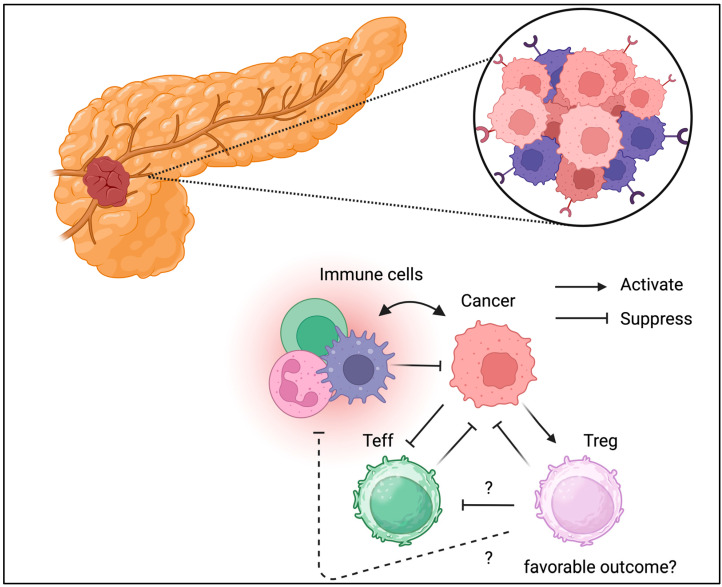
**Tregs can be detected in TME of pancreatic cancer.** Unlike the other cancer types, Tregs might be associated with favorable outcomes in pancreatic cancer.

**Figure 7 biomedicines-13-01173-f007:**
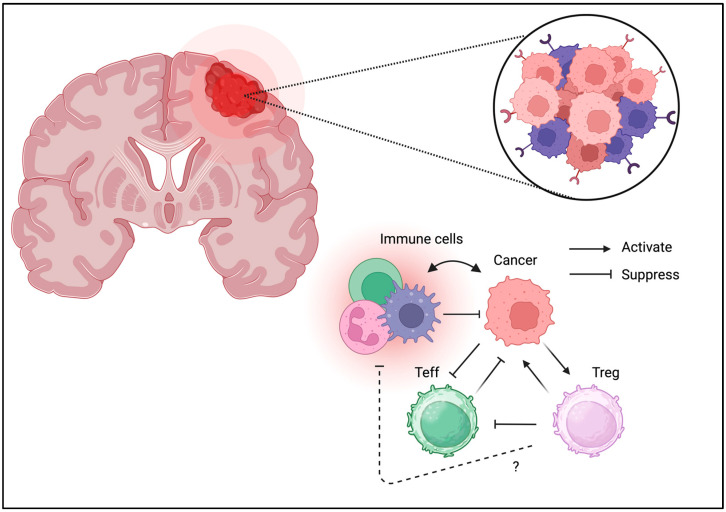
**Tregs can be detected in TME of brain tumors.** Similarly to the other cancer types, Tregs may be associated with unfavorable outcomes in brain tumors.

**Table 1 biomedicines-13-01173-t001:** Core and additional Treg markers.

Phenotypic Markers	Increased/Decreased in Tregs	Description
CD4	Increased	Core Treg markers
CD25	Increased	Core Treg markers
CD127	Decreased	Core Treg markers
FOXP3	Increased	Core Treg markers
CTLA-4	Increased	Additional Treg markers
GARP	Increased	Additional Treg markers
GITR	Increased	Additional Treg markers
IKZF2	Increased	Additional Treg markers
OX40	Increased	Additional Treg markers
CD69	Increased	Additional Treg markers
TIGIT	Increased	Additional Treg markers
ICOS	Increased	Additional Treg markers
LAG-3	Increased	Additional Treg markers
PD-1	Increased	Additional Treg markers
CD39	Increased	Additional Treg markers
CD73	Increased	Additional Treg markers

**Table 2 biomedicines-13-01173-t002:** Phenotype and prognosis of cancer infiltrating Tregs.

Cancer Types	Phenotype	Prognosis
Colorectal cancer	FOXP3 low	Favorable
Lung cancer	CCR8+	Unfavorable
Breast cancer	CCR8+	Unfavorable
Pancreatic cancer	CCR5+	Favorable
Hepatocellular carcinoma	Unknown	Unknown
Brain tumor	Unknown	Unfavorable
Kidney/Renal cell carcinoma	CCR4+ (Prostate cancer)	Unknown
Melanoma	Unknown	Unfavorable
Gynecological cancer	CCR10 (Ovarian cancer)	Unknown
Head and Neck cancer	FOXP3 hi	Unfavorable
Soft tissue sarcoma	Unknown	Unknown
Leukemia/Lymphoma	Unknown	Unknown

## Data Availability

Not applicable.

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
