# Peer review of "The Role of Tregs in the Tumor Microenvironment"

_biomedicines, 2025, doi:10.3390/biomedicines13051173_

Round 1
Reviewer 1 Report
Comments and Suggestions for Authors
This review provides a broad overview of the role of Tregs in the tumor microenvironment across various cancer types. The manuscript is well-structured, with a logical flow from the biology of Tregs to their cancer-specific roles and therapeutic implications. It offers a valuable perspective for researchers interested in cancer immunotherapy and tumor immunology. However, several issues should be addressed to enhance its scientific rigor, clarity, and utility:
1. The manuscript contains numerous grammatical errors (e.g., "antidoby" instead of "antibody", "understating" instead of "understanding", "heterogenisity" instead of "heterogeneity").
2. Section 3.2 would benefit from a more comprehensive discussion of the immune microenvironment and Tregs. I recommend to include recent studies such as PMID: 38433921, highlights how Treg contribute to immune modulation in lung cancer.
3. I recommend revising the manuscript to italicize the word "gene" wherever it refers to the gene name, in accordance with the conventions for scientific writing.
4. Section 3.3 discussing the role of Tregs in breast cancer would benefit from the inclusion of recent multi-omic studies. For instance, PMID: 39252976 provides valuable insights into how metabolic alterations may intersect with immune regulation, including Treg function.
5. In the Section 3.4, the manuscript would benefit from the inclusion of recent researchs. For example, PMID: 37667257 demonstrates comprehensive insights into the Treg and TME in Pancreatic cancer.
6. To enhance the discussion of tumor microenvironmental complexity in gynecological cancers (Section 3.9), I suggest to cite PMID: 38278958, which provides comprehensive insights into the Treg and TME in gynecological cancer.
Author Response
This review provides a broad overview of the role of Tregs in the tumor microenvironment across various cancer types. The manuscript is well-structured, with a logical flow from the biology of Tregs to their cancer-specific roles and therapeutic implications. It offers a valuable perspective for researchers interested in cancer immunotherapy and tumor immunology. However, several issues should be addressed to enhance its scientific rigor, clarity, and utility:
- The manuscript contains numerous grammatical errors (e.g., "antidoby" instead of "antibody", "understating" instead of "understanding", "heterogenisity" instead of "heterogeneity").
(Response) I would like to apologize for the grammatical errors. The manuscript has been revised by professional language service (editage).
- Section 3.2 would benefit from a more comprehensive discussion of the immune microenvironment and Tregs. I recommend to include recent studies such as PMID: 38433921, highlights how Treg contribute to immune modulation in lung cancer.
(Response) I would like to appreciate the suggestions. The article was included in the text.
- I recommend revising the manuscript to italicize the word "gene" wherever it refers to the gene name, in accordance with the conventions for scientific writing.
(Response) The manuscript was revised accordingly.
- Section 3.3 discussing the role of Tregs in breast cancer would benefit from the inclusion of recent multi-omic studies. For instance, PMID: 39252976 provides valuable insights into how metabolic alterations may intersect with immune regulation, including Treg function.
(Response) I would like to appreciate the suggestions. The article was included in the text.
- In the Section 3.4, the manuscript would benefit from the inclusion of recent researchs. For example, PMID: 37667257 demonstrates comprehensive insights into the Treg and TME in Pancreatic cancer.
(Response) I would like to appreciate the suggestions. The article was included in the text.
- To enhance the discussion of tumor microenvironmental complexity in gynecological cancers (Section 3.9), I suggest to cite PMID: 38278958, which provides comprehensive insights into the Treg and TME in gynecological cancer.
(Response) I would like to appreciate the suggestions. The article was included in the text.
Reviewer 2 Report
Comments and Suggestions for Authors
This review article addresses the important and timely topic of regulatory T cell (Treg) function within the tumor microenvironment (TME) and their impact on cancer development, progression, and metastasis. The author correctly emphasizes that TME represents a complex ecosystem comprising extracellular matrix, immune cells, stromal components, and vasculature, which plays a crucial role in tumor immune evasion. Particular attention is given to the dual nature of Tregs - while they promote immunosuppression and cancer progression, they also exhibit heterogeneity and plasticity, making their role tumor-type dependent.
In line 22, the word "petide" should be corrected to "peptide", and the text should be carefully proofread for other typos.
Section 2.1 on Treg phenotype and function needs better structuring. I recommend clearly distinguishing between core Treg markers (such as FOXP3, CD25, low CD127 expression) and additional markers that may vary depending on tumor type or microenvironment conditions (CTLA-4, LAG-3, GITR).
Despite promising preclinical data, targeted therapies against Tregs have not yet yielded significant clinical success. The article would benefit from deeper analysis of potential reasons for this translational gap.
The addition of such critical analysis would significantly strengthen the review's scientific value and clinical relevance.
Author Response
This review article addresses the important and timely topic of regulatory T cell (Treg) function within the tumor microenvironment (TME) and their impact on cancer development, progression, and metastasis. The author correctly emphasizes that TME represents a complex ecosystem comprising extracellular matrix, immune cells, stromal components, and vasculature, which plays a crucial role in tumor immune evasion. Particular attention is given to the dual nature of Tregs - while they promote immunosuppression and cancer progression, they also exhibit heterogeneity and plasticity, making their role tumor-type dependent.
In line 22, the word "petide" should be corrected to "peptide", and the text should be carefully proofread for other typos.
(Response) I would like to apologize for the grammatical errors. The manuscript has been revised by professional language service (editage).
Section 2.1 on Treg phenotype and function needs better structuring. I recommend clearly distinguishing between core Treg markers (such as FOXP3, CD25, low CD127 expression) and additional markers that may vary depending on tumor type or microenvironment conditions (CTLA-4, LAG-3, GITR).
(Response) I fully agreed with the reviewer. It must be more informative bona fide Tregs markers/additional functional markers.
Despite promising preclinical data, targeted therapies against Tregs have not yet yielded significant clinical success. The article would benefit from deeper analysis of potential reasons for this translational gap. The addition of such critical analysis would significantly strengthen the review's scientific value and clinical relevance.
(Response) I fully agreed with the reviewer. This topic should be discussed for the future clinical development. We have provided following sentences in the section.
“The presence of heterogeneity in the cancer types, patient conditions including immunity, other non-immune cells may contribute to the limited efficacy of TME-directed therapy in the current time point. Further understanding of both TME and immune cells including Tregs may beneficial to discover TME-directed therapeutic approaches.”
Reviewer 3 Report
Comments and Suggestions for Authors
In this review, the author aims to define the role of regulatory T cells in TME, given the role of phenotypic plasticity and heterogeneity of Treg, as well as Treg-targeted therapeutic approaches. The manuscript covers a broad range of sometimes controversial information in the context of Treg functioning in TME. However, the work may benefit from addressing some points.
Some general concerns:
In all paragraphs devoted to the cancer types the Treg phenotype should be placed where it is possible. Maybe the table could be added reflecting Treg phenotypical characteristics in specific cancer types.
Some figures (2-6) in my opinion may be combined into one, because they represent similar things.
Captions for figures should be added.
It is necessary to follow the text and give full names before using abbreviations (line 22 FOXP3, line 37 and others)
Other concerns:
Line 46: typo – heterogeneity
Part 2 lines 60-62: I suggest revealing the correlation between FoxP3 and Treg functioning. Alternative splicing of FoxP3 in cancer may play a role in the existence of Tregs with various phenotypic and functional characteristics
In part 2.1 the author aims to review phenotype and function of Tregs. It would be great to clarify the role of each mentioned marker in Treg biology, perhaps in the form of a table.
Line 81: typo – heterogeneity
Part 2.2. In accordance with the title of this part the author should at first define the difference between plasticity and heterogeneity of Tregs. The difference is not clear for the reader
Line 94: typo – heterogeneous
Part 2.3 may be improved by adding the phenotype of tissue-resident Tregs. The author may also reflect this in Figure 1. What characterizes Tregs in the CNS, skin, etc.?
Line 104: typo - tissue-resident
Part 3. I think the association between Tregs and unfavorable outcome should be reviewed in a separate paragraph. Given that such association characterizes many cancer types, author could describe the exact mechanism by which Tregs functioning in the TME leads to tumor immune evasion (include association of Tregs with other immune cells, etc). Maybe the author could add a scheme
Line 116: reference is missing
Line 122: ref is missing
Lines 128-129: In which cases the infiltration of Tregs associated with better prognosis? What makes Tregs not suppressive in these cases? Describe the phenotype?
Line 143: What is meant by divergent phenotypes? The explanation should be provided
Line 177: «unique phenotype» which?
Lines 177-178: Association between immunotherapy in triple-negative breast cancer and Tregs in TME is not clear. What is immunotherapy aimed at?
Paragraph 3.3. Lines 209-213: This part is devoted to the role of Tregs in type 1 diabetes. Is this information applicable for a review on Tregs in TME?
Line 218: immunotherapy aimed at?
Line 264: what characterizes dysfunctional Tregs? Phenotypical characterization is missing
Line 326: what immune checkpoints were blocked?
Line 328: «several cancers» which?
Lines 350-351: PD-1 and PD-L1 on which cells?
Lines 365-366: «It is hypothesized that leukemic stem cell» The sentence seems to be not finished
Part 4. Development of therapeutic approaches targeting Tregs in TME is a field of extensive research. The author should expand this part and more deeply review the investigated approaches. Which other Treg-associated molecules are being involved in therapy?
Line 385, 399: typo – drug
Line 412: typo - however
Author Response
In this review, the author aims to define the role of regulatory T cells in TME, given the role of phenotypic plasticity and heterogeneity of Treg, as well as Treg-targeted therapeutic approaches. The manuscript covers a broad range of sometimes controversial information in the context of Treg functioning in TME. However, the work may benefit from addressing some points.
Some general concerns:
In all paragraphs devoted to the cancer types the Treg phenotype should be placed where it is possible. Maybe the table could be added reflecting Treg phenotypical characteristics in specific cancer types.
(Response) I fully agreed with the idea. The table of Treg phenotype and characteristics in specific cancer types would be informative.
Some figures (2-6) in my opinion may be combined into one, because they represent similar things.
(Response) It may be possible to combine them, however, the phenotype and interaction of Tregs is not identical across different cancer cell types.
Captions for figures should be added.
(Response) I have added figure captions.
It is necessary to follow the text and give full names before using abbreviations (line 22 FOXP3, line 37 and others)
(Response) I spelled out the gene names as suggested.
Other concerns:
Line 46: typo – heterogeneity
(Response) The typo was corrected.
Part 2 lines 60-62: I suggest revealing the correlation between FoxP3 and Treg functioning. Alternative splicing of FoxP3 in cancer may play a role in the existence of Tregs with various phenotypic and functional characteristics
(Response) This is an important point, however, it was not possible to find the literature describing isoform expression with regards to the cancer.
In part 2.1 the author aims to review phenotype and function of Tregs. It would be great to clarify the role of each mentioned marker in Treg biology, perhaps in the form of a table.
(Response) Thank you so much for the suggestion. It would be informative to provide the each Treg marker in the table.
Line 81: typo – heterogeneity
(Response) The typo was corrected.
Part 2.2. In accordance with the title of this part the author should at first define the difference between plasticity and heterogeneity of Tregs. The difference is not clear for the reader
(Response) Thank you so much for your suggestion. We have distinguished plasticity and heterogeneity.
Line 94: typo – heterogeneous
(Response) The typo was corrected.
Part 2.3 may be improved by adding the phenotype of tissue-resident Tregs. The author may also reflect this in Figure 1. What characterizes Tregs in the CNS, skin, etc.?
(Response) Thank you so much for your suggestion. Unlike conventional TRM, the phenotype of tissue resident Tregs were not fully identified.
Line 104: typo - tissue-resident
(Response) The typo was corrected.
Part 3. I think the association between Tregs and unfavorable outcome should be reviewed in a separate paragraph. Given that such association characterizes many cancer types, author could describe the exact mechanism by which Tregs functioning in the TME leads to tumor immune evasion (include association of Tregs with other immune cells, etc). Maybe the author could add a scheme
(Response) We have added a figure showing the dual role of Treg in cancer progression together with the literatures.
Line 116: reference is missing
(Response) The reference (Whiteside, 2014 Semin Cancer Biol) was added.
Line 122: ref is missing
(Response) The reference (Weiden et al. 2017 Nat Rev Immunol) was added.
Lines 128-129: In which cases the infiltration of Tregs associated with better prognosis? What makes Tregs not suppressive in these cases? Describe the phenotype?
(Response) We provided description as follows.
“Tregs, characterized by reduced FOXP3 expression, may not be suppressive and enhance local immune reactions [51].”
Line 143: What is meant by divergent phenotypes? The explanation should be provided
(Response) We provided description as follows.
“In addition to their divergent phenotypes such as co-inhibitory molecule expression, Tregs in breast cancer may contribute to disease progression (Figure 4) [62].”
Line 177: «unique phenotype» which?
(Response) We provided description as follows.
“Tregs isolated from breast cancer cells exhibit a unique phenotype including CCR8 expression [60].”
Lines 177-178: Association between immunotherapy in triple-negative breast cancer and Tregs in TME is not clear. What is immunotherapy aimed at?
(Response) We provided description as follows.
“Recently, immunotherapy such as PD-1/PD-L1 immune checkpoint blockade has been used to manage triple-negative breast cancer, suggesting the involvement of Tregs in the TME [61].”
Paragraph 3.3. Lines 209-213: This part is devoted to the role of Tregs in type 1 diabetes. Is this information applicable for a review on Tregs in TME?
(Response) I would like to explain the importance of tissue resident Treg in local immune controls. I still think it may be informative to address the role of Treg in T1D.
Line 218: immunotherapy aimed at?
(Response) We provided description as follows.
“Immunotherapy, such as PD-1 and CTLA-4 checkpoint blockade, has been indicated for pancreatic cancer management; however, its efficacy has not been fully confirmed, partially because of disease severity [79].”
Line 264: what characterizes dysfunctional Tregs? Phenotypical characterization is missing
(Response) According to the literature, IL-17 producing Treg may promote liver fibrosis. We provided description as follows.
“Tregs reside in the liver tissue, and liver Treg dysfunction such as IL-17 production is associated with fibrosis [84].”
Line 326: what immune checkpoints were blocked?
(Response) We provided description as follows.
“Immune checkpoint blockade including PD-1 and CTLA-4 blockade has also been attempted in cervical cancer treatment [116].”
Line 328: «several cancers» which?
(Response) We provided description as follows.
“The roles of the TME and Tregs in gynecological cancers are not fully understood, although immunotherapy has been attempted for several cancers including ovarian, cervical and uterine cancers.”
Lines 350-351: PD-1 and PD-L1 on which cells?
(Response) We provided description as follows.
“Limited efficacy of immunotherapies such as PD-1 and PD-L1 blockade for soft tissue sarcoma has been reported [122].”
Lines 365-366: «It is hypothesized that leukemic stem cell» The sentence seems to be not finished
(Response) We provided description as follows.
“Similarly to HSC, Leukemic stem cells (LSCs) have also been supported by Treg [132].”
Part 4. Development of therapeutic approaches targeting Tregs in TME is a field of extensive research. The author should expand this part and more deeply review the investigated approaches. Which other Treg-associated molecules are being involved in therapy?
(Response) This topic should be discussed for the future clinical development. We have provided following sentences in the section.
“The presence of heterogeneity in the cancer types, patient conditions including immunity, other non-immune cells may contribute to the limited efficacy of TME-directed therapy in the current time point. Further understanding of both TME and immune cells including Tregs may beneficial to discover TME-directed therapeutic approaches.”
Line 385, 399: typo – drug
(Response) The typo was corrected.
Line 412: typo - however
(Response) The typo was corrected.
Round 2
Reviewer 1 Report
Comments and Suggestions for Authors
Authors have made substantial improvements to the manuscript. The revised version is more robust and interpreted. I appreciate the expanded discussion of potential biological mechanisms. Here are some minor Suggestions:
1. The manuscript contains numerous grammatical errors.
Line 22 "FOXP3 inhibiting petide" → "FOXP3 inhibiting peptide"
Line 82 "heterogeniety" → "heterogeneity"
Line 115 "memotry" → "memory"
2. The supported reference regarding the role of Tregs in gynecological cancer appears somewhat insufficient. It is recommended to cite recent literature, such as PMID: 39360160.
3. It would also be beneficial to incorporate findings from the latest scRNA-seq or spatial transcriptomics studies and propose future directions for "precision Treg-targeted interventions."
4. Additionally, is author from Jike University or Jikei University?
Author Response
Authors have made substantial improvements to the manuscript. The revised version is more robust and interpreted. I appreciate the expanded discussion of potential biological mechanisms. Here are some minor Suggestions:
- The manuscript contains numerous grammatical errors.
Line 22 "FOXP3 inhibiting petide" → "FOXP3 inhibiting peptide"
Line 82 "heterogeniety" → "heterogeneity"
Line 115 "memotry" → "memory"
(Response) The quality of language editing was very poor. I would like to aplogize. The typo has been corrected.
- The supported reference regarding the role of Tregs in gynecological cancer appears somewhat insufficient. It is recommended to cite recent literature, such as PMID: 39360160.
(Response) We have checked the suggested reference, however, the reference is not relevant to the immune reactions. Therefore, it was not possible to include the reference, but we would like to appreciate the suggestion.
- It would also be beneficial to incorporate findings from the latest scRNA-seq or spatial transcriptomics studies and propose future directions for "precision Treg-targeted interventions."
(Response) We provided the statements in the discussion as follows.
“Further understanding of both TME and immune cells including Tregs may beneficial to discover TME-directed therapeutic approaches. However, this may not be readily possible due to patient sample limitations and ethical issues. Recent scRNA-seq or spatial transcriptomics studies may provide further insights with regards to immune profile of TMEs. The discovery of the crosstalk between cells in the TME may lead to novel therapeutic interventions. “
- Additionally, is author from Jike University or Jikei University?
(Response) The quality of language editing was very poor. I would like to aplogize. Jikei University (NOT Jike University)
Reviewer 2 Report
Comments and Suggestions for Authors
I am satisfied with the author's response.
Author Response
Thank you so much for your appreciation. Your comments improved the quality of the manuscript.
Reviewer 3 Report
Comments and Suggestions for Authors
Thank you for addressing my concerns.
However, two concerns remain
1. The authors did not address the comment about the impact of alternative splicing of FoxP3. Please indicate what splice variants are found or could be generated in Treg and what are their potential implications in cancer.
2. Figure legends. Please provide a brief explanation immediately following the figure title.
Author Response
Thank you for addressing my concerns.
(Response) Your insightful and thoughtful comments improved the quality of the manuscript. I have fully agreed with your previous and current scientific suggestions.
However, two concerns remain
1. The authors did not address the comment about the impact of alternative splicing of FoxP3. Please indicate what splice variants are found or could be generated in Treg and what are their potential implications in cancer.
(Response) We have provided the statement as follows.
“In mice, Tregs were initially identified as CD4+ and CD25+ cells [20]. FOXP3 has been identified as the master regulator [16]. CD4(+)CD25(+)CD127(low/-) cells express FOXP3, and FOXP3+ Tregs can be isolated without FOXP3 staining [21]. In human, Tregs are known to express FOXP3 delta-2 isoform lacking exon 2. The role of FOXP3 delta-2 isoform is not fully confirmed in the context of TME.”
Figure legends. Please provide a brief explanation immediately following the figure title.
(Response) We have provided the description of the figure.